# Multiparametric Sonographic Imaging of Thyroid Lesions: Chances of B-Mode, Elastography and CEUS in Relation to Preoperative Histopathology

**DOI:** 10.3390/cancers14194745

**Published:** 2022-09-29

**Authors:** Moritz Brandenstein, Isabel Wiesinger, Julian Künzel, Matthias Hornung, Christian Stroszczynski, Ernst-Michael Jung

**Affiliations:** 1Institute of Diagnostic Radiology, Interdisciplinary Ultrasound Department, University Hospital, 93053 Regensburg, Germany; 2Department of Otorhinolaryngology, Head and Neck Surgery, University Hospital of Regensburg, 93053 Regensburg, Germany; 3Department of Surgery, University Hospital, 93053 Regensburg, Germany

**Keywords:** solid thyroid lesions, B-mode, shear-wave elastography, CEUS, histopathology

## Abstract

**Simple Summary:**

As the incidence of thyroid lesions in Europe is rising, more and more people affected by thyroid pathologies seek treatment in a clinic. Every suspicious thyroid nodule needs to be confirmed as benign or malignant in order to be treated correctly. Unnecessary invasive diagnostics and thyroid surgery should be avoided. The aim of this retrospective study was to improve the distinction between benign and malignant nodules by using new high-performance multiparametric ultrasound examination techniques. By analyzing 122 thyroid nodules we created a score-based system combining B-mode, shear-wave elastography and contrast-enhanced ultrasound malignancy criteria. This system allows for a quite accurate detection of thyroid carcinomas with a sensitivity of 95% and specificity of 75.49%. Shear-wave elastography and contrast-enhanced ultrasound can detect unique malignancy features, which cannot be found in B-mode. Therefore, these criteria would present a relevant addition to the B-mode TI-RADS classification.

**Abstract:**

Background: The aim was to improve preoperative diagnostics of solid non-cystic thyroid lesions by using new high-performance multiparametric ultrasound examination techniques. Methods: Multiparametric ultrasound consists of B-mode, shear-wave elastography and contrast enhanced ultrasound (CEUS) including Time-Intensity-Curve (TIC) analysis. A bolus of 1–2.4 mL Sulfur Hexafluorid microbubbles was injected for CEUS. Postoperative histopathology was the diagnostic gold standard. Results: 116 patients were included in this study. 102 benign thyroid nodules were diagnosed as well as 20 carcinomas. Suspicious B-mode findings like microcalcifications, a blurry edge and no homogeneous sonomorphological structure were detected in 60, 75 and 80% of all carcinomas but only in 13.7, 36.3 and 46.1% of all benign lesions. The average shear-wave elastography measurements of malignant lesions (4.6 m/s or 69.8 kPa centrally and 4.2 m/s or 60.1 kPa marginally) exceed the values of benign nodules. Suspicious CEUS findings like a not-homogeneous wash-in and a wash-out were detected almost twice as often in carcinomas. Conclusion: Multiparametric ultrasound offers new possibilities for the preoperative distinction between benign and malignant thyroid nodules. A score based system of B-mode, shear-wave and CEUS malignancy criteria shows promising results in the detection of thyroid carcinomas. It reaches a sensitivity of 95% and specificity of 75.49%.

## 1. Introduction

There are several factors, which increase the probability of developing struma nodosa: iodine-deficient areas, the female sex as well as increasing age. The risks for malignant thyroid pathologies are manifold and include radiation, long-term exposure to metals as well as first degree relatives with thyroid cancer [1,2]. Ultrasound is globally the best available diagnostic tool for evaluating thyroid pathologies. Several B-mode TI-RADS criteria help by hinting at malignancy, but they lack contrast enhanced ultrasound (CEUS) perfusion. Furthermore they only recommend elastography, instead of including it into its classification [3,4]. Unfortunately, an invasive Fine needle aspiration (FNA) or postoperative histopathology is still required for a more accurate clarification of thyroid lesions as benign or malignant [5].

CEUS detects further benignity or malignancy indicators: The pattern of the contrast-agent expansion called wash-in and the later pattern of its disappearance called wash-out. These allow for an evaluation of the dynamic microvascularization of a lesion. Time-intensity-curves (TICs) create quantified detailed information about the uptake, distribution and disappearance of the contrast agent [6,7,8,9,10].

The EFSUMB guidelines state that CEUS is most powerful when combined with other conventional sonographic techniques like B-mode and elastography. Therefore and due to the increasing thyroid malignancy incidence in Europe, performing CEUS on thyroid nodules is an active high potential research field. Unfortunately, its incorporation into the everyday clinical practice is not recommended, yet [9].

The aim of this study was to use modern multimodal ultrasound for a more precise preoperative characterization of thyroid lesions before undergoing a biopsy. Thus, unnecessary thyroid surgery and FNAs should be avoided. We tried to close the sensitivity gap between EU-TI-RADS (78–88%) and cytology (76–98%) [11,12]. Therefore, different sonographic findings in B-mode, shear-wave elastography and CEUS including TICs were rated according to their single and combined probability of predicting a thyroid lesion as benign or malignant.

## 2. Materials and Methods

The study was approved by the local ethics committee (20-2122-104). Every patient signed a declaration of consent before starting CEUS. The indication for inclusion in this study were the same as for the operative (hemi-)thyroidectomy: lymph node metastases with possible origin in thyroid gland, symptoms like difficulties in breathing and swallowing, elevated Calcitonin tumor marker in blood count, suspicious preliminary sonographic examinations leading to a TI-RADS 4 or 5 rating or a TI-RADS 3 lesion appearing as a cold node in scintigraphy.

### 2.1. B-Mode and Elastography Examination Technique

116 patients with suspicious non-cystic and solid thyroid lesions were examined on a high-performance ultrasound device (Ge Logiq E9; GE, Solingen, Germany). We useda multifrequency probe (Ge 9L-D Linear with 2.0 to 9.0 MHz). The examinations were documented with single pictures or Cine-Loops saved as digital DICOM files. The size of every thyroid gland along with its nodules was measured in three dimensions. The morphology of the lesion in B-mode was assessed using EU-TI-RADS criteria: structure of lesion-margins, microcalcifications and homogeneity [3].

For rating the echogenicity in B-mode, a scale from 1 to 3 was defined: 1 is used for a complete lack of echogenicity (echofree), 1.5 for lower echogenicity (hypoechoic), 2 for the same echogenicity as the surrounding thyroid tissue (isoechoic), 2.5 and 3 for higher echogenicity (hyperechoic).

Afterward the stiffness of the nodule was quantified using shear-wave elastography. Therefore, 8 regions of interest (ROIs) were placed across the lesion: 2 in the center, 4 alongside the margins and 2 in the normal thyroid tissue for comparison. Then the values were merged into one central, one marginal and one external measurement series. If the lesion was not big enough for 6 ROIs, only one central and one marginal ROI were placed across it. The stiffness was quantified in [m/s] and [kPa].

### 2.2. CEUS Examination Technique

For CEUS 1–2.4 mL Sulfur Hexafluorid microbubbles (Sonovue, Bracco; Milan, Italy) along with 10 mL sodium chloride were injected into a cubital vein. During the CEUS examination one recording was created digitally as a Cine-loop consisting of three phases: the early arterial phase 10 to 15 s after the bolus injection, the late arterial phase 30 s later and the venous phase up to one minute after the injection. For wash-out detection late recordings were created up to 5 min after the bolus.

The parametric colour-coded CEUS mode visualized the timing of the expansion of the contrast agent within the nodule. A scale from 0 to 4 was created in order to translate the colours into numbers (Table 1):

During every CEUS examination a TIC analysis was performed with focus on the time to peak (TTP) [s]. Therefore, 8 ROIs were placed across the lesion and merged similar to the ones of shear-wave elastography. Within every ROI the changes of the contrast agent intensity were calculated and depicted as a TIC visualizing the wash-in and wash-out phenomenon. TTP is defined as the time interval in seconds between bolus injection of the contrast agent and the point of peak contrast enhancement [13].

### 2.3. Statistical Analysis

The statistical analyses were performed with SPSS 25.0 (SPSS Inc., Chicago, IL, USA). The Kruskal–Wallis test compared the measured values of the benign to the malignant nodules and calculated the statistical significance. Probabilities less than 0.05 were considered statistically significant. The ROC-curve and the Youden’s J statistic calculated cut-off values with a high sensitivity and specificity for elastography (m/s and kPa) and TTP (seconds). Binary logistic regression provided the diagnostic accuracy and DOR for every examination technique.

## 3. Results

116 patients (49 males, 67 females; age 24–84 years; mean 56 ± 14 years) were included in this study. Inter- or postoperative histopathological analysis proved 102 benign thyroid nodules (57 ± 13 years) and 20 carcinomas (50 ± 13 years). Histopathology identified 11 carcinomas as papillary, 4 as follicular, 4 as medullary and one as an anaplastic thyroid carcinoma. Every carcinoma was removed via thyroidectomy.

The diameters of the benign regressive thyroid nodules ranged from 6 to 104 mm (mean 26 ± 17 mm) and the carcinoma diameter from 8 to 150 mm (mean 28 ± 32 mm). The size of thyroid lesions could be a relevant factor for its malignancy with carcinomas having a slightly higher average value. However, it should be noted that their standard deviation is almost twice as high as in benign ones.

### 3.1. B-Mode Findings

Compared to the benign thyroid lesions, 34% more carcinomas had no homogeneous sonomorphological structure, as shown in Table 2. Furthermore, the carcinomas showed a blurry edge more than twice and microcalcifications more than four times as frequently as benign nodules. The microcalcifications were the rarest finding to occur in benign lesions with only 13.7%. Examples for these B-mode findings are depicted in Figure 1.

The echogenicity interquartile range of the benign nodules reaches from 1.5 to 2 whereas the carcinomas range from 1 to 1.5 both centrally and marginally. The malignant lesions show the lowest echogenicity values along their margins with a median of 1. The center and margin of benign lesions as well as the center of malignomas score a median of 1.5. Overall, the carcinomas present themselves with a lower average echogenicity than the benign nodules.There were significant differences between the two groups in the center (*p* = 0.002) and along the margins (*p* = 0.00011).

### 3.2. Shear-Wave Elastography

Table 3 summarizes the results of shear-wave elastography measurements. The carcinomas surpass the benign lesions in their center and along their margins in both m/s and kPa absolutely and relatively. The two groups differ statistically significantly in their centers (*p* = 0.002) and along their margins (*p* = 0.004). The highest average shear-wave elastography values of 4.6 m/s and 69.8 kPa as well as the highest standard deviation were measured in the malignant lesion’s center. Figure 2 depicts the shear-wave elastography examination of a malignant nodule with hard margins. Figure 3 shows a soft benign lesion.

### 3.3. Contrast-Enhanced Ultrasound (CEUS) Findings

#### 3.3.1. Wash-In Dynamics

Centrally both benign and malignant lesions share similar rather slow wash-in dynamics with a median of 2 and an interquartile range from 1 to 3. There is no statistically significant difference in this category (*p* = 0.97).

Along their margins benign and malignant nodules differ statistically significantly (*p* = 0.000001). The majority of lesions show higher values marginally representing a faster wash-in. The highest values and thus the fastest wash-in was captured along the carcinomas’ margins with a median score of 3.5 and an interquartile range from 3 to 4.

#### 3.3.2. (In-)Homogeneous Wash-In and Wash-Out

In CEUS a not-homogeneous wash-in and a partial or complete wash-out occurred more than twice as often in malignant lesions than in benign ones. Wash-out is the malignant CEUS finding spotted most frequently with 85% of all examined carcinomas. The results are presented in Table 4. Figure 4 depicts the typical wash-in and wash-out dynamics of a carcinoma and Figure 5 of a benign lesion in CEUS.

#### 3.3.3. Time-Intensity-Curve (TIC) Analysis

Table 5 summarizes the results of TIC analysis measurements focusing on the TTP [s]. There were statistically significant differences between benign and malignant lesions in both TTP measurement series (*p* = 0.004 marginally and *p* = 0.003 centrally). The carcinomas peak approximately 6 s ahead of the benign nodules centrally and marginally. 13.2 s as the fastest average TTP was scored by the center of carcinomas. The relative TTP values of benign and malignant nodules were similar (approximately 90%). Figure 6 depicts the TIC-analysis of a malignant lesion.

### 3.4. Diagnostic Accuracy, Diagnostic Odds Ratio, Sensitivity, Specificity, Positive and Negative Predictive Value

The diagnostic accuracy, DOR, sensitivity and specificity of every examination technique are listed in Table 6. The highest values for diagnostic accuracy were scored by shear-wave elastography: especially by the measurements along the lesions’ margins with 90.8%. Fast marginal wash-in reached the highest DOR of 21.3. The highest sensitivity value was scored by a low echoic margin and center of the lesion with 95.0% and 90.0%. A faster central and marginal wash-in compared to the surrounding thyroid tissue reached the highest specificity value with 96.8% and 96.9%, followed by partial or complete wash-out with 95.2%. Malignancy criteria with both high values for sensitivity and specificity were: partial or complete wash-out, microcalcifications as well as fast marginal wash-in.

## 4. Discussion

Several multicenter studies, which examined more than 10,000 patients, confirmed the following malignancy criteria: blurry edge demarcation, not-homogeneous sonomorphological structure, microcalcifications and a low echogenicity in B-mode; stiff regions with elevated shear-wave elastography values; a not-homogeneous fast wash-in as well as a partial or complete wash-out in CEUS [3,7,8,9,10,14,15].

According to several reviews the average sensitivity values for the B-mode malignancy criteria microcalcifications, hypoechogenicity, blurry edge and inhomogeneous structure are: 40 to 54%, 63 to 73%, 51 to 56% and 48%. Multicenter studies also reported the following average DOR for these B-mode findings: 6.8 for microcalcifications, 4.5 to 5 for hypoechogenicity and 6.1 to 6.9 for a blurry edge demarcation [8,16]. The sensitivity values calculated in this study all surpass the reported values by 20 to 30%. Our DOR for microcalcifications and hypoechogenicity exceed the values of multicenter studies whereas the blurry edge has a lower ratio. A possible explanation therefore would be that there is no unambiguous definition for these four B-mode criteria except of microcalcifications.

In case of shear-wave elastography most clinical studies discuss cut-off values for carcinomas of 2.5 m/s or 30 kPa. The sensitivity for these published cut-off values ranged from 75% to 92%. The average diagnostic accuracy was 85% [14,16,17,18]. This study rendered limit stiffness values of approximately 4 m/s and 50 kPa with a sensitivity of 75% and a diagnostic accuracy of 90%. These results adequately reflect the findings of multicenter studies. Unfortunately, shear-wave cut-offs cannot be easily compared between disparate devices (especially of different manufacturers). This is due to anatomical, operator-dependent and technological differences [19].

The typical CEUS dynamics of a malignant lesion are a fast irregular expansion of the contrast agent in combination with a wash-out, from late arterial to venous phase. Both wash-in and wash-out curves of thyroid carcinomas run lower than in healthy thyroid tissue [10]. CEUS as a whole has an average sensitivity of 85% to 90%, diagnostic accuracy of 85% and DOR of 52 according to meta-analyses [4,7,8,14,20]. The diagnostic accuracy of fast marginal wash-in, not-homogeneous wash-in and partial or complete wash-out as single malignancy criteria are 81%, 90% and 82% [9,21,22]. In comparison, these numbers correspond well to those of this study.

The fast marginal wash-in is the only CEUS malignancy criterion to exceed the values from literature in diagnostic accuracy. Furthermore, its DOR of 21.3 is the top score of this study. However, there were also some adenomas, which showed a partial wash-out.

In a study about TIC perfusion analysis researchers reported the following average TTP values: 8.7 s along carcinomas’ margins and 8.9 s in their centers as well as 13.9 s marginally and 7.7 s centrally in adenomas [23,24]. In comparison, the average numbers of our study exceed the others by 5 to 12 s. Furthermore, our carcinomas peak significantly earlier than the benign lesions, suggesting a faster and greater contrast agent uptake in malignomas than in benign nodules.

We tried several combinations of malignancy criteria for a more precise differentiation of thyroid lesions. A sequence of fast marginal wash-in and wash-out points almost clearly at carcinomas with a sensitivity of 100% and a specificity of 64%. If neither microcaclifications nor a fast marginal wash-in can be detected, you are most likely dealing with a benign nodule. This combination reaches a sensitivity of 75% and a specificity of 85%. If the decision for a thyroidectomy had relied on these two criteria, only 16 instead of 74 out of 96 patients with benign nodules had undergone surgery. This would mean a decrease of 60%. Furthermore, we created a score consisting of 3 (to 4) B-mode, shear-wave elastography and CEUS malignancy criteria: non-homogeneous sonomorphological structure; marginal shear-wave elastography measurements above the cut-off value of 4.0 m/s or 50.7 kPa; non-homogeneous wash-in (and wash-out). Every fulfilled criterion adds one point to the score. If a lesion is rated with 2 points or more, it has a high risk of malignancy. This score-based system reaches a sensitivity of 95% and specificity of 75.49% (100% and 72.55% for 4 malignancy criteria).

Unfortunately, using multiparametric ultrasound did not lead up to a final differentiation of a thyroid lesion as benign or malignant in all cases. Further multimodal sonographic studies are required to determinate certain contrast conduction and hemodynamic patterns as well as better elastography cut-off values. Nevertheless, the sensitivity, specificity, diagnostic accuracy and DOR of all sonographic imaging modes, especially CEUS, have significantly improved in recent years [10,20,24,25]. These modern multiparametric malignant findings represent a relevant addition to the B-mode TI-RADS criteria. Additionally, comparing all malignancy criteria we noticed the following fact: marginal categories always had higher values for sensitivity, specificity and diagnostic odds ratio than central ones. For future shear-wave elastography and TIC-analysis studies, placing 2 ROIs marginally and 1 centrally across the thyroid lesion instead of 6 should be sufficient.

Most studies perform histological diagnostics by FNA or sonographic biopsy [10,14,21]. Nevertheless, the intra- or postoperative histopathological examination used in our study is considered as equal.

Possible limitations for this study are: its retrospective approach and the time consuming, on high-end-technology depending examination. There are no officially standardized schemes for CEUS and TIC-procedures [25,26]. In addition, the examinations had to be performed by an experienced sonographer on a preliminary examined group of patients. Although having a higher sensitivity than EU-TI-RADS, neither FNA nor a postoperative histopathology can always determine a thyroid nodule as 100% benign or malignant [11,12].

## 5. Conclusions

In conclusion, the modern, extensive, multiparametric sonographic diagnostics seem to achieve promising results in the differentiation between benign and malignant thyroid nodules. Thus, the number of unnecessary thyroid surgery in case of benign lesions can be reduced by more than 50%. Shear-wave and CEUS features would make a relevant addition to the TI-RADS classification, which relies only on B-mode. In the course of this, marginal findings appear to be more significant than central ones. Nevertheless, further multicenter studies are required for standardizing the qualitative and quantitative sonographical examination techniques and features as well as their classification.

## Figures and Tables

**Figure 1 cancers-14-04745-f001:**
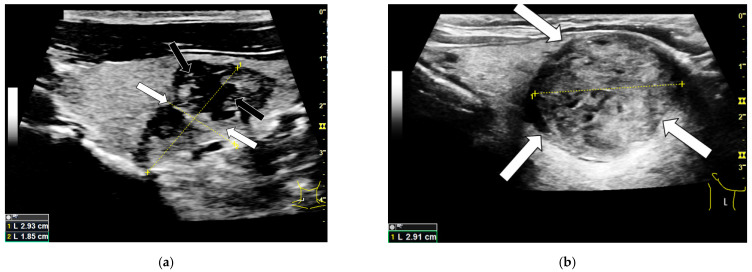
B-mode examination of a right thyroid lobe with histo-pathologically proven thyroid carcinoma (**a**) and a histo-pathologically proven thyroid adenoma (**b**). The carcinoma shows typical malignancy criteria like a blurry edge demarcation (white arrows) as well as an irregular sonomorphological structure and microcalcifications (black arrows). Additionally the majority of the malignoma can be described as hypoechoic compared to the surrounding thyroid tissue. The adenoma shows a clear edge demarcation (white arrows) and a quite homogeneous structure compared to the carcinoma. It is partly isoechoic and partly hyperechoic.

**Figure 2 cancers-14-04745-f002:**
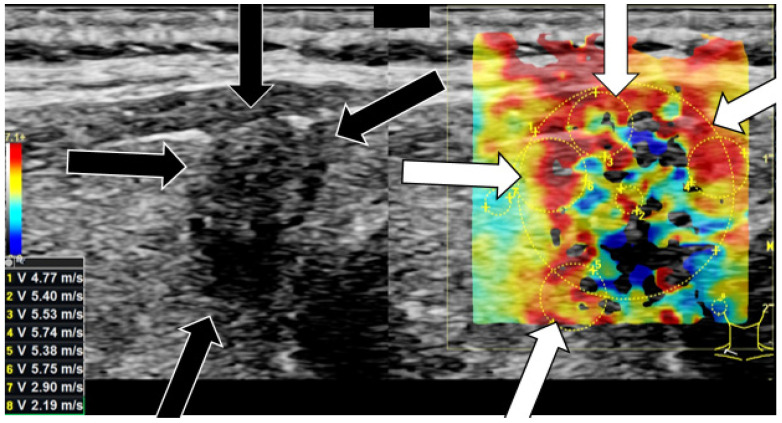
Shear-wave elastography [m/s] examination on a right thyroid lobe with thyroid carcinoma. The arrows (black arrow in B-mode; white arrow in shear-wave elastography) point at irregular hard areas. The stiffness values of the lesion all surpass the calculated cut-off value of 4.04 marginally and 3.88 centrally. The stiffness values of the normal thyroid tissue are lower.

**Figure 3 cancers-14-04745-f003:**
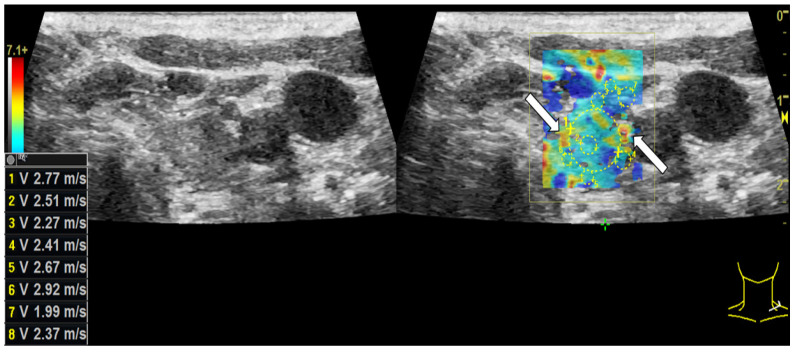
Shear-wave elastography [m/s] examination on a left thyroid lobe with thyroid adenoma. Only few hard areas (white arrows) can be detected in shear-wave elastography. The major part of the nodule appears rather soft. Neither the stiffness values of the lesion nor of the normal thyroid tissue surpass the calculated cut-off value of 4.04 marginally and 3.88 centrally.

**Figure 4 cancers-14-04745-f004:**
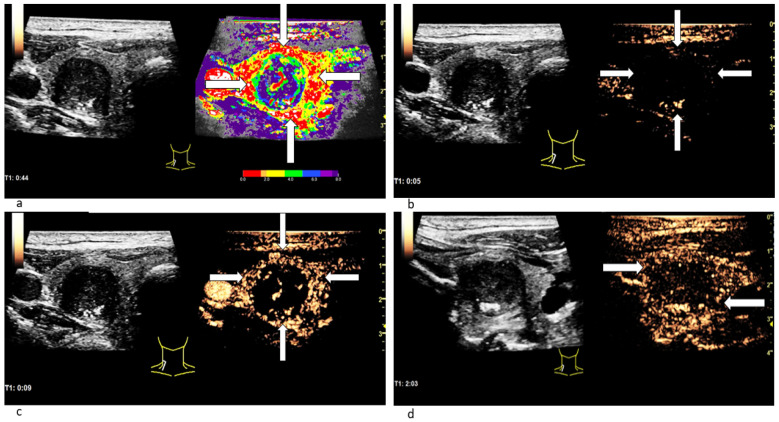
Patient with thyroid carcinoma. In the parametric colour-coded CEUS mode (**a**) the margins of the lesion (white arrows) are mainly covered in red and orange. The center of the lesion as well as the normal thyroid gland are primarily coloured in purple with small parts of blue and green. This suggests that SonoVue first expands along the carcinoma’s margins. The CEUS examination 5 s after the bolus injection (**b**) shows no contrast agent expansion yet neither marginally (white arrows) nor centrally. 9 s after the bolus injection (**c**) the contrast enhancement has begun: Microbubbles first expand along the lesion’s margins (white arrows). In the CEUS recording, taken 2 min after bolus injection (**d**), a decrease of contrast enhancement called wash-out along the carcinoma’s margins (white arrows) can be spotted.

**Figure 5 cancers-14-04745-f005:**
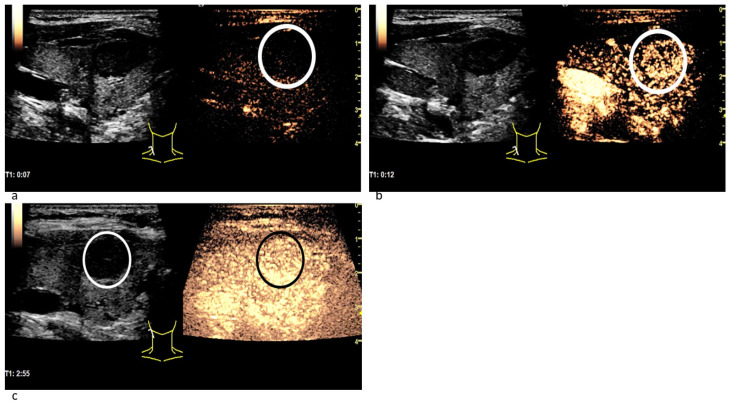
Patient with a benign thyroid nodule. The CEUS examination 7 s after the bolus injection (**a**) shows no contrast agent expansion in the nodule (white circle), yet. 12 s after the bolus injection (**b**) the contrast enhancement has begun: Microbubbles expand inside the lesion quite homogeneously (white circle) and not from the margins to the center. In the CEUS recording taken almost 3 min after bolus injection (**c**) the contrast agent intensity has not decreased: neither in the nodule (white circle in B-mode; black circle in CEUS) nor in the surrounding thyroid tissue. No wash-out phenomenon has occurred.

**Figure 6 cancers-14-04745-f006:**
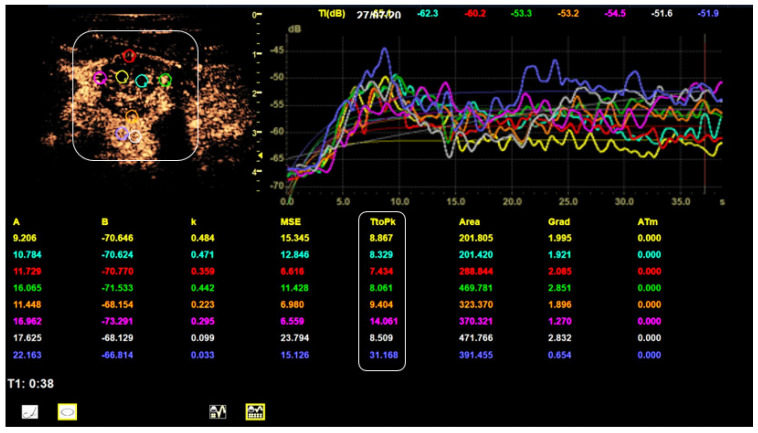
TIC analysis of a right thyroid lobe with thyroid carcinoma. The smoothed TICs for every ROI are depicted in the upper right corner. The measured TTP values (bottom white box) meet the calculated malignancy criteria cut-off values: The 2 central values are lower than 13.5 s and the 3 out 4 marginal values are lower than 13.1 s.

**Table 1 cancers-14-04745-t001:** Colour-coded contrast enhanced ultrasound (CEUS) microbubble expansion classification.

Contrast Agent Expansion Dynamics (In a Thyroid Nodule Compared to Healthy Thyroid Tissue)	Colour	Number
Very late	Blue/Purple/Black	0
Late	Green	1
Slightly late	Yellow	2
Simultaneously	Orange	3
Fast	Red	4

**Table 2 cancers-14-04745-t002:** Frequency of malignant B-mode findings in benign and malignant thyroid nodules.

B-Mode Findings	Benign	Malignant
blurry edge demarcation	36.3%	75%
microcalcifications	13.7%	60%
inhomogeneous sonomorphological structure	46.1%	80%

**Table 3 cancers-14-04745-t003:** Average absolute and relative shear-wave elastography stiffness quantifications in m/s and kPa of benign and malignant thyroid nodules: in their center, along their margins and in the surrounding thyroid tissue.

Shear-Wave Elastography Measurements	Benign (*n* ± STD)	Benign (%)	Malignant (*n* ± STD)	Malignant (%)
center [m/s]	3.50 ± 1.0	160.34%	4.60 ± 1.3	226.56%
margin [m/s]	3.49 ± 0.7	160.39%	4.22 ± 1.1	207.74%
surrounding tissue [m/s]	2.18 ± 0.8	100.00%	2.03 ± 0.7	100.00%
center [kPa]	41.16 ± 25.5	260.3%	69.80 ± 44.8	499.61%
margin [kPa]	40.19 ± 14.8	254.15%	60.09 ± 25.1	430.16%
surrounding tissue [kPa]	15.81 ± 12.5	100.00%	13.97 ± 6.6	100.00%

**Table 4 cancers-14-04745-t004:** Frequency of malignant CEUS findings in benign and malignant thyroid nodules.

CEUS Findings	Benign	Malignant
inhomogeneous wash-in	28.4%	75%
partial/complete wash-out	37.9%	85%

**Table 5 cancers-14-04745-t005:** Average absolute and relative time to peak (TTP) [s] measurements of benign and malignant thyroid nodules: in their center, along their margins and in the surrounding thyroid tissue.

TIC Analysis Mean	Benign (*n* ± STD)	Benign (%)	Malignant (*n* ± STD)	Malignant (%)
TTP [s] center	19.7 ± 10.8	91.0%	13.2 ± 6.5	89.0%
TTP [s] margin	19.9 ± 8.7	91.8%	13.9 ± 6.5	93.7%
TTP [s] surrounding tissue	21.7 ± 15.1	100.0%	14.8 ± 7.6	100.0%

The cut-off values for TTP [s] are listed in Table 6.

**Table 6 cancers-14-04745-t006:** Diagnostic accuracy, DOR, sensitivity and specificity for every examination technique.

Examination Technique		Diagnostic Accuracy (%)	Diagnostic Odds Ratio	Sens. (%)	Spec. (%)
B-mode	blurry edge demarcation	83.6	2.5	75.0	63.7
	microcaclifications	83.6	9.4	60.0	86.3
	inhomogeneous sonomorphological structure	83.6	4.7	80.0	53.9
	marginal hypoechogenicity	83.6	11.8	95.0	46.1
	central hypoechogenicity	83.6	5.5	90.0	41.2
Shear-wave elasto-graphy	marginal cut-off: 4.0 m/s	90.0	3.5	75.0	81.4
	central cut-off: 3.9 m/s	88.4	2.3	75.0	70.1
	marginal cut-off: 50.7 kPa	90.8	1.1	75.0	75.3
	central cut-off: 46.9 kPa	89.2	1.0	75.0	72.2
CEUS	fast marginal wash-in	88.7	21.3	50.0	96.9
	fast central wash-in	82.6	0.9	20.0	96.8
	inhomogeneous wash-in	82.6	7.2	34.9	93.1
	partial/complete wash-out	82.6	3.4	66.7	95.2
CEUS: TIC	marginal TTP cut-off: 13.1 s	78.5	1.1	57.9	81.1
	central TTP cut-off: 13.5 s	78.5	1.1	78.9	73.0

## Data Availability

The data was acquired during everyday clinical examinations in the University Clinic Regensburg. The data presented in this study are available upon request from Co-author and head of the Ultrasound Department of University Clinic Regensburg EM Jung.

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
