# Peer review of "Multiparametric Sonographic Imaging of Thyroid Lesions: Chances of B-Mode, Elastography and CEUS in Relation to Preoperative Histopathology"

_cancers, 2022, doi:10.3390/cancers14194745_

Round 1
Reviewer 1 Report
The study by Brandenstein M. et al. aims to develop more effective methods of distinction between benign and malignant thyroid nodules by combining B-mode, elastography and CEUS. According to the authors a score-based system based on the results of these measurements allows for accurate detection of thyroid carcinomas (with a sensitivity of 95% and specificity of 75,49%).
The subject of this paper is relevant and manuscript is well-written, presented in an intelligible fashion and the language is clear and correct. All procedures have been clearly described (with very good graphical presentation). The statistical methods used are well chosen. References are up to date and appropriate.
The only minor issue is the relatively small number of analyzed cases (114), however, for the purposes of this study, this is sufficient to draw the correct conclusions.
In my opinion, this manuscript is suitable for publication in Cancers.
Author Response
Thank you for your comments.
I added a conclusion paragraph, changed the referenced to MDPI style and reduced long sentences.
Reviewer 2 Report
It requires some changes in spelling and grammar.
In addition, I would like to see more examples to contrast positive vs. negative findings.
Considering the high volume of daily studies, how practical is it to do this exam in real life?
Author Response
Thank you for your comments.
I revised spelling and grammar, reduced long sentences, added a conclusion, changed the references to MDPI style and added more examples to contrast positive vs negative findings (shear-wave as well as wash-in and wash-out with pictures)